# The Role of Ion Channels and Chemokines in Cancer Growth and Metastasis: A Proposed Mode of Action Using Peptides in Cancer Therapy

**DOI:** 10.3390/cancers16081531

**Published:** 2024-04-17

**Authors:** Gerald J. Mizejewski

**Affiliations:** 1Division of Translational Medicine, Molecular Diagnostics Laboratory, Albany, NY 12201, USA; gerald.mizejewski@health.ny.gov; Tel.: +518-486-5900; Fax: +518-402-5002; 2Wadsworth Center, New York State Department of Health, Empire State Plaza, Albany, NY 12201, USA; 3Biggs Laboratory, Empire State Plaza, Albany, NY 12237, USA

**Keywords:** alpha-fetoprotein, metastasis, cell cycle, breast cancer, chemokines, calcium, peptides, therapy

## Abstract

**Simple Summary:**

Cancer Metastasis has been mostly misunderstood and underestimated by the general public regarding cancer deaths. The aims and objectives of the present report was to more fully describe the scientific activities and components that are involved in the malignant cell’s fulfillment of the metastatic process and to impress the readership that metastasis is the major cause on all cancer deaths. The paper proceeds to enumerate and describe certain factors that contribute to cancer cell proliferation and subsequent metastasis such as; (a) calcium levels, (b) multiple cell membrane channels, and (c) the chemokine/receptor system. These latter components could serve to provide ideal molecular targets for possible future peptide therapeutic applications in treating cancer patients.

**Abstract:**

Metastasis (Met) largely contributes to the major cause of cancer deaths throughout the world, rather than the growth of the tumor mass itself. The present report brings together several of the pertinent contributors to cancer growth and metastatic processes from an activity standpoint. Such biological activities include the following: (1) cell adherence and detachment; (2) cell-to-cell contact; (3) contact inhibition; (4) the cell interfacing with the extracellular matrix (ECM); (5) tumor cell-to-stroma communication networks; (6) chemotaxis; and (7) cell membrane potential. Moreover, additional biochemical factors that contribute to cancer growth and metastasis have been shown to comprise the following: (a) calcium levels in the extracellular matrix and in intracellular compartments; (b) cation voltage and ATP-regulated potassium channels; (c) selective and non-selective cation channels; and (d) chemokines (cytokines) and their receptors, such as CXCL12 (SDF-1) and its receptor/binding partner, CXCR4. These latter molecular components represent a promising group of an interacting and synchronized set of candidates ideal for peptide therapeutic targeting for cancer growth and metastasis. Such peptides can be obtained from naturally occurring proteins such as alpha-fetoprotein (AFP), an onco-fetal protein and clinical biomarker.

## 1. Introduction: Cancer Growth and Metastasis

### 1.1. The Process of Cancer Cell Growth and Proliferation

Cancer is viewed as a condition of the uncontrolled growth of cells/tissues as a result of gene/chromosome instability caused by DNA mutations [1]. One of the most remarkable discoveries in cancer growth research within the past several decades has been the regulation of cell division, growth, and proliferation by means of a biological clock within cells termed the “growth cell cycle” [2]. The activities of this built-in cellular clock take place within five phase periods (namely, G1, S, G2, mitosis, and the resting stage, G0). One phase called mitosis is the period in which cell division occurs. In order to regulate the cell cycle (i.e., cell growth), modulating factors called cyclin-dependent kinase enzymes (CdKs) and their protein inhibitors serve as enhancers, regulators, and checkpoint suppressors, respectively; these factors advance or halt progression at each phase cycle. Certain peptides (see below) can interfere and block cell cycle progression at the G1-to-S-phase transition, a prequel to a DNA synthesis phase. With the S-phase cell cycle arrested, the cell fails to advance to the mitosis stage in which cell division occurs; hence, cell growth and proliferation do not occur. The suppression of the cell growth cycle is an ideal means to prevent cancer growth. It is of further interest that the FDA recently approved cell cycle inhibitors of cell cycle CdKs for human use in cancer therapy [3]. The mechanism of peptide cell cycle cessation involves the presence of CdK cell cycle inhibitors, p27 Kip1 and p21 Cip1, by preventing their degradation, i.e., breakdown. It is also through such inhibitory factors that certain peptides can prevent the attachment of a phosphate group (phosphorylation) to a cell cycle suppressor termed p53 in order to regulate its activity. This latter step prevents subsequent cell cycle progression, hence forthcoming cell growth. 

As a final note, peptides can target cancer cells, unlike most, if not all, chemotherapeutic agents in use today. Peptides, due to their amphipathic properties, can seek out and selectively bind to cancer cells versus normal cells due to an aberrant electrical charge difference on the cell surface bilayer of cancer cells [4]. Peptides also do not exhibit off-target homing specificity. Peptides can then bind and penetrate into the interior of the cancer cell to eventually cause cell cycle arrest [5]. As an additional benefit, peptides are ideally suited for use as post-surgical cancer agents in combinations with standard chemotherapy drugs; the ancillary action could prevent the re-growth of recently removed tumor masses from the surgical excision area. 

### 1.2. The Process of Metastasis

The vast majority of cancer deaths can be attributed to breakaway tumor cells (metastasized) which migrate to distant organ sites, rather than demise from the primary tumor mass itself. It has been demonstrated in previous reports that metastasis-related proteins can participate in cell membrane activities such as tumor cell adhesion and contact with the extracellular matrix (ECM) [6,7]. Such associations further indicate that growth factors and their receptors together with various kinases are involved and interact with MET cells in order to achieve the completed process of cancer cell detachment and migration from the main tumor mass [8]. Once tumor cells are separated and disseminated from their tumor mass, detached freed tumor cells begin to migrate and spread. The tumor cells proceed to traverse various ECM layers and basement membranes in order to migrate and traverse nearby blood vessels. Following the involvement of cancer cell adhesion and attachment sites, proteolytic enzymes from the cells are concomitantly released and activated to digest a passageway (path) through a myriad of tissue/cell membrane barriers. The tumor cells must then gain entrance into and traverse the endothelial cells of capillaries and the lumen of blood vessels and/or lymphatic ducts [4]. The tumor cells’ extravasation and subsequent migration to distant tissue sites can only occur after their exit from the blood vessel’s lumen [9]. 

Once inside the capillary lumen, circulating tumor cells seek to attach themselves to already formed clusters of aggregated platelets that themselves have adhered to the inner wall of blood vessels. This process of platelet attachment is referred to as “tumor cell-induced platelet aggregation [10]”. This endothelial attachment site provides circulating tumor cells with a temporary “shielded haven” while circulating through the blood vessels to distant homing sites. By means of a process called “chemical-induced chemotaxis,” tumor cells are attracted to distal homing sites, usually directed to filtration organs such as the liver, lungs, kidneys, and bone marrow [11]. By means of this chemoattraction, sensitized tumor cells emerge through the blood vessel walls and are drawn to the direction of specific organ/tissue destinations by cytokine agents called “chemokines” and their respective receptors [12,13]. This system of dual attraction chemo-agents (i.e., CXCL12 chemokine and the CXCR4 receptor) is then directed to where the tumor cells are destined to become nested. This is within the stromal tissue of the predestined specific host organ/tissue site [8,14]. Thus, the guidance of metastatic tumor cells to distal organ sites is a complicated multi-step process involving a pair of protein cytokine families comprising (A) chemokines and (B) their matched binding receptors’ partners [14].

## 2. Alpha-Fetoprotein Contains a Peptide Fragment That Can Suppress Cancer Growth and Metastasis

Human alpha-fetoprotein (HAFP) has a long history of clinical use as a tumor-associated fetal biomarker, utilized to detect both fetal defects during pregnancy and cancers in adults [15]. HAFP is a single-chain polypeptide with an average molecular mass of 69 kDa, depending on its carbohydrate micro-heterogeneity [16]. This fetal protein has been classified as a member of the albuminoid gene family, consisting of AFP, albumin, vitamin D-binding protein, and the AFP-related (ARG) protein [13]. HAFP binds to a vast array of ligands, including various drugs, dyes, steroid hormones, heavy metals, flavonoids, fatty acids, and phytoestrogens [17].

Human AFP has been reported to contain an encrypted “occult” hidden peptide fragment (named Growth Inhibitory Peptide or GIP) concealed within the compacted tertiary folded HAFP polypeptide [18]. This AFP-derived peptide fragment is exposed following a conformational change in the U-shaped full-length native AFP molecule. The GIP fragment of AFP is a 34 amino acid (AA) peptide which was identified and purified as a segment from AAs #445 to 480 of the human AFP polypeptide chain [19,20]. GIP has been synthesized as a free peptide fragment and subsequently isolated, purified, and biochemically characterized; its biological activities were then determined [21]. GIP was found to inhibit and suppress growth in both estrogen (E)-dependent and non-E-dependent (basal) cancer growth and can interact and bind with various transcription factors, dyes, and cell cycle-associated proteins [22,23,24]. The mechanism of action of GIP as a cell cycle suppressor has also been established and reported; GIP has been further associated with cell membrane penetration and disruption and can suppress both cancer cell growth and migration [19,21].

GIP is an amphipathic peptide with an isoelectric point of pH 4.7; it displays a secondary structure consisting of 45% beta sheets and turns, 45% random coil (disordered), and 10% alpha-helix [19,20]. Hence, GIP is largely a beta-sheet peptide and contains a carboxyterminal type-1 reverse beta-turn structure which serves to enhance cell surface membrane binding. The AFP-derived peptide is a cell membrane disrupter which lacks a receptor and bypasses receptor-mediated endocytosis. Nonetheless, GIP undergoes rapid cell internalization via a channel and/or pore forming processes [25,26]. Following transmembrane passage, the peptide becomes diffusely distributed throughout the cytoplasm within 1.0 h and by 2.0 h, is localized in a focused fashion in the perinuclear region of the cell, an area contiguous with the endoplasmic reticulum [21]. The GIP segment is bioactive both as a linear as well as a cyclized (disulfide bridge) peptide [27,28,29]. 

AFP-derived GIP displays multiple biological activities including estrogen-associated cytoskeletal interactions and the prevention of birth defects in animal models [20,30]. Plasma cell membrane electrical activities in the GIP segment have also been reported by means of electrophysiological procedures such as the following: (a) sharp microelectrodes and (b) patch-clamp whole-cell recordings in cultured MCF-7 cell membranes [31]. In vitro data suggested that GIP at low concentrations could serve as a “cell channel-forming” peptide coincident with a decreased cell membrane electrical resistance ([32,33,34,35,36], and personal communication). At higher peptide concentrations, GIP was capable of cell membrane pore forming activities with a cell membrane potential at −30 to −45 mV ([5,25,26]; see acknowledgement). Thus, GIP could act either as a channel-forming or a pore-forming peptide coincident with an increased cell membrane electrical resistance. Thus, the AFP-derived peptide was found to gain entrance into cells via (a) cell membrane bilayer pore formation and/or (b) a channel formation action both of which were concentration-dependent [5]. In addition, a global RNA microarray analysis of GIP administrated to MCF-7 breast cancer cell cultures for 8 days demonstrated that GIP was capable of the up- and/or down-regulation of the RNA of multiple outward/inward flux cation channel proteins resulting in cation channel regulation and cell cycle arrest in breast cancer cell cultures [5] (Table 1, Part I and Part II).

The cytoplasmic ripple effect of membrane ion channel regulation via GIP has been reported to affect and halt G1-to-S-phase cell cycle progression demonstrated by growth inhibition in MCF-7 human breast cancer cells [32,33,34,37,38] (Table 1 Section 1). An RNA microarray data set (see above) revealed that the treatment of MCF-7 cells with GIP for 8 days resulted in the RNA down-regulation of multiple cell cycle proteins such as Cyclin-E, SKP2, and associated transition checkpoint proteins which prevented Cyclin-E/Cdc2-induced G1-to-S-phase progression [5]. GIP concomitantly blocked the ubiquitin-induced degradation of cyclin inhibitors such as p27 KIP and p21 CIP resulting in cell cycle arrest and subsequent mitotic cell growth inhibition [5]. While the labile linear version of GIP forms timers which are growth-inhibitory at high molar concentrations, the cyclic form (which behaved like a dimer) was inhibitory at lower molar concentrations. Moreover, both forms were cytostatic without any cytotoxic side effects in cell cultures (in vitro) and in xenograft mouse models via in vivo studies [19,20].

Expression of 716 transcripts was significantly altered in MCF-7 cell after 8 days of treatment with GIP as compared to treatment with scrambled peptide. Four hundred and thirty RNAs were down-regulated, while 286 RNAs were up-regulated; ** = real-time PCR. Data provided by Kathleen Arcaro, University of Massachusetts, Amherst, MA.

The disruption of cancer cell signaling activities has been shown to disturb, impair, and disable the ability of tumor cells to transduce downstream signals in order to adhere and spread. GIP has been described as a cell surface membrane perturbation agent, capable of disrupting tumor cell adhesion, membrane pseudopodial extensions, cell shape, aggregation, and cell agglutinating activities [19,23,24]. Thus, peptides exemplified by GIP might have the potential to serve as a cancer therapeutic agent alone or as an adjunct chemotherapeutic agent for cancer.

## 3. Membrane Ion Channels, Growth/Proliferation, and Metastasis: Cell Cycle Involvement

Membrane ion channels are essential for cell proliferation and may have a role in cancer progression and its subsequent metastasis. Microarray data for such roles have been confirmed for potassium voltage-gated channels as well as Ca^++^ and other cation-regulated channels ([5]; Table 1). Some of these channels include voltage-gated selective potassium-dependent channels in addition to non-selective calcium, magnesium, and sodium (TRP) channels. A cell cycle-dependent linkage has been further demonstrated between ion channels and the cell growth cycle [32,33]. Other membrane channels can also affect the membrane voltage and intracellular Ca^2+^ signal transduction in cell growth [28]. Homeostatic parameters, such as intracellular ion concentration, cytosolic pH, cell volume, and cell cycle regulatory agents, are governed by the action of cation channels. Furthermore, environmental factors, such as low oxygen pressure, acidosis, and exposure to various serum proteins, can affect physiological processes such as growth, proliferation, and metastasis [34]. The homeostatic ratio of the extra- and intracellular content of Ca^2+^ levels has further been shown to influence platelet aggregation, a process required for the adhesion of circulating metastatic tumor cells to the interior of blood vascular walls [10,36]. In summary, voltage-gated and non-selective cation channel activity may significantly contribute to both growth/proliferation and metastatic processes. Interestingly, alpha-fetoprotein-derived GIP displays amino acid sequence identities/similarities to a vast array of cation channel-associated and calcium-interacting proteins suggesting a possible mimetic action [36] (see Table 2).

The metastatic ability of cancer cells has been found to involve the expression of various ion channel activities in strongly versus weakly metastatic tumor cell lines [39]. Such differences in cells related to their extracellular versus intracellular Ca^2+^ levels have especially been denoted in cancer cells. Differences in the voltage-independent non-selective Ca^2+^ influx channels (transient receptor potential (TRP) channels) and the voltage-dependent K^+^-regulated channels were further sought out and studied. Both cell-to-cell contact-mediated cancer cell migration and growth are controlled by multiple sensory stimuli of TRP channels from stresses such as the following: pain, heat, cold, peppermint oil (menthol), camphor, capsaicin (hot peppers), mustard, Wasabi, ginger, touch, pressure, vibrations, and mechanical motion [37,40]. Overall, such TRP channels appear to affect cell migration by the regulation and transport of extracellular Ca^2+^ ions activated by spice and other sensory-activated agents. Ca^2+^ channels, together with serotonin receptors, are also important for tumor cell lodgment (nesting) into target organ/tissue sites as the endpoint of metastatic nesting. Interestingly, serotonin is one of the more important platelet-released secretions that also influences tumor cell lodgment into tissue distal “nesting” sites [41]. Cell surface membrane negative changes have also been found to be important for the cell migration of the metastatic cell membrane’s electrical potential [42]. It is the cell membrane phospholipid components (phosphodiesterase, phosphatidylcholine, phosphotidylserine) that are also crucial for determining the overall cell surface charge (membrane flip) of a cell membrane bilayer [43].

Potassium (K) channels are known to function in the cell membrane of cells to open and close channels by means of the following: (1) membrane voltage; (2) ligand binding to juxtaposed G-coupled receptors; (3) changes in intra- and extracellular calcium concentrations; (4) the alteration of cell membrane tension; and (5) changes in cell membrane electrical resistance and (6) ATP-sensitive currents [44]. K channels are also known to regulate progression through the cell cycle and to contribute to cell growth, proliferation, cell-produced extensions, motility, invasion, and metastasis [45]. In contributing to the metastatic process, K channels can also modulate cell proliferation and migration, apoptosis, cell adhesion, Ca^2+^ influx to the cytoplasm, and cytoskeletal alterations in the microtubular system [20].

In a global microarray analysis of GIP’s effect on MCF-7 cultured breast cancer cells [5], it was reported that GIP was able to down-regulate multiple types of potassium- and calcium-associated channels (Table 1). The cation channels affected by GIP regulation consisted of the following types: (1) potassium voltage-gated channels (KCNB2); (2) potassium voltage-gated channel KQT-like (KCNQ3); (3) calcium channel, voltage-dependent-2 (CACNA2D4); (4) calcium channel, voltage-dependent (CACNC6); (5) transient receptor potential (TRP)-V4 (Valanoid family); and (6) transient receptor potential (TRP)-M8 (Melastatin family) [40]. As shown in Table 1, (Section 2), the RNA GIP microarray results showed that the voltage-gated potassium and calcium channels in MCF-7 breast cancer cells were RNA down-regulated from 1.8 to 8.0-fold decreases [5].

## 4. The Role of Calcium in Cation Channel Regulation

Calcium is a universal cell second messenger, regulating many important processes such as growth, differentiation, and programmed cell death. All cells in the body are bathed in extracellular fluids very rich in Ca^2+^, while intracellular cytoplasmic Ca^2+^ concentrations were found to be 3 logs lower in concentration [46,47]. However, normal cell membranes are impermeable to Ca^2+^ influx and require ion channels to enter cells. Within the cell, calcium ions are stored in the smooth endoplasmic reticulum (ER) and are pumped out of the ER; the disruption of the cell membrane acts to disable cell surface permeability with a consequent influx of Ca^2+^ ions [48]. Calcium ions are biologically very active, being capable of a considerable disturbance of the metabolic order resulting in a large intracellular toxic accumulation of Ca^2+^ ions. However, high levels of cytoplasmic ions can result in calcium toxicity to a cell [49,50]. 

It has been established that Ca^2+^ ions play a major role in toxic cell injury especially involving cell membrane disruption and leakage. In prior reports, Ca^2+^ ions have been demonstrated to be the mediators of plasma membrane disruptive damage and their functional consequences thereof [49]. Thus, detrimental alterations in Ca^2+^ homeostasis have been documented as major factors in mechanisms leading to lethal plasma membrane damage, which transforms a living cell into a necrotic one.

In light of the role that calcium can play in cell death and the composition of the TRP calcium channel signal complex, an examination of the data was elegantly reported by Nilius and Wondergem [34]. Such investigations were eventually applied to studies of GIP action on LNCAP prostate cancer cells (personal communications and see Acknowledgements). These electrophysiological (grant-funded) studies indicated that GIP (A) increased the electrical conductance (inward current due to calcium influx) across the cell membrane and (B) decreased membrane resistance. Such studies and others using menthol spice activation revealed that Ca^2+^-activated K^+^ ion (BK) channels were also involved; further studies showed that GIP enhanced the TRP valloniid calcium channel rather than inhibiting it ([40] and personal communication/acknowledgement). Therefore, it may be postulated that GIP may not block calcium entry but actually promote an increased flow (influx) of calcium into cultured MCF-7 cells. TRP channels are transiently open indicating that some of the total TRP channels are open much of the time. However, it can be proposed that GIP could open many of the channels at a given moment and maintain the open state for a period of time, such that high amounts of extracellular calcium might enter into the cancer cell. If GIP were to be administered constantly over several days (i.e., 8 days), it might be assumed that a number of cells would gradually be destroyed from calcium “toxic” overload [40]. Thus, GIP might at first appear to be cytostatic, but over an extended period of time (8 days), GIP treatment might ultimately lead to cell death via Ca^2+^ toxicity. 

## 5. Properties of Chemokines concerning Metastasis

Chemokines are chemical attractant cytokines that are locally produced in tissues and traffic the emigration of lymphoid cells from the blood stream into sites of inflammation and/or growth/proliferation [51]. Chemokines can direct and influence many physiological processes such as angiogenesis, inflammation, degranulation, cell proliferation, autoimmunity, HIV infection, tumor growth, parasite infections, and leukocyte trafficking [11]. Discovered only decades ago, chemokines are now known to function as “local regulatory” molecules in leukocyte (and other cells) maturation, trafficking, homing, and the progressive development of lymphoid and other tissues. Chemokines are known to comprise a family of 50 ligands and 20 receptors, which, despite their size, are remarkably homogenous with properties similar to Interleukin-8 (IL-8), the first chemokine to be discovered [11]. The effect of chemokines on leukocytes (and other cells) are mediated by hepahelical G-coupled receptors that form a link to GTP-binding proteins. Many chemokines are produced as a result of pathological conditions, some fulfill housekeeping (regulatory) roles, and some can function during histo- and organogenesis [11]. Chemokines attracted world-wide attention when it was discovered that some of their receptors function as binding sites for AIDS viruses (HIV) on macrophages and that AFP binds to those macrophage surface receptors. In fact, AFP was found to bind to the CXCR5 receptor [52]. Although they show receptor binding with HIV agents, chemokines/receptors’ main functions are attracting, homing, regulating, and guiding cells into distal tissues.

Chemokine CXCL12 is the bone marrow stromal cell-derived factor-1 (SDF-1) which modulates cell migration, proliferation, and regulates growth in stromal tissues [52]. The SDF-1 chemokine binds to the G-coupled receptor CXCR4 and can activate multiple signal transduction pathways such as in the mitogenic kinase system [12,53]. In fact, SDF-1α activates the p44/42 mitogen-activated protein kinases termed ErK-1 and ErK-2 and the mitogen-activated protein kinase-kinases [54]. SDF-1 also activates phosphatidylinositol 3-kinase, increases NF-kB activity in cells, and up-regulates the adhesion focal tyrosine in addition to paxillin, a focal adaptor scaffolding protein [55]. Although conceivable, it has not been reported, to date, that GIP could actually bind the CXCR4 receptor (see below).

By means of Genebank sequence matching, it was discovered that GIP appears to resemble a portion of chemokine molecules (see Table 3), especially the SDF-1 stromal cell growth chemokine (Table 3). It could be further proposed from direct and indirect evidence that GIP appears to possess the properties of a chemokine segment or be a chemokine-associated peptide (Table 4) [43]. If such is the case, logic would dictate that GIP could bind to three possible chemokine cell surface receptors, i.e., CCR5, CXCR4, or CCR6. The CCR5 receptor has already been reported to bind the full-length HAFP molecule and, hence, would be a good candidate. CXCR6 is the only receptor found in reproductive tissues as well as lymphoid and bone marrow tissue [13,56]. Thus, CXCR4 is the remaining receptor deemed possible to bind to GIP. CCR6 receptors are also known to bind to antimicrobial cell-penetrating peptides, and GIP possesses properties of an antimicrobial (pore-forming) peptide [57]. Since there is no receptor reported for GIP, it could provide an additional mechanism for GIP’s action of growth suppression since GIP might serve as a decoy ligand to compete for binding to the SDF-1 chemokine or to its receptor, CXCR4 [43,58].

If indeed GIP serves as a decoy ligand (posing as a mimic of SDF-1), GIP could possibly bind to the CXCR4 receptor of the G-coupled family but not activate the receptor to induce cell proliferation (Table 3). For example, full-length HAFP has been reported to bind to the CCR5 receptor [59]. This, in effect, might cause suppression in growth factor-stimulated cells such as prostate and breast cancers, which have actually been reported. Also, it might be possible that GIP could bind to a possible chemokine receptor, induce the cell shape change, induce tubulin (or actin) polymerization, and also inhibit cell migration (spreading), cell adhesion, and cell proliferation as reported for GIP [20]. In other words, GIP may be a partial antagonist or a chemokine mimic depending on its concentration (dose) and length of treatment duration (Table 3 and Table 4). Thus, GIP has an advantage over the other chemokine-related inhibitors which cannot act in a partial antagonist fashion. It can also be proposed that GIP might present as a novel type of chemokine inhibitor or decoy ligand, in that it could still retain the growth suppressive property and anti-angiogenic capability. GIP could still be able to induce some of the early (initial) functions of a true chemokine (such as immunostimulation). Finally, it was reported that GIP was capable of inhibiting metastasis in 60% of tumors and in an in vivo study involving a GI-101 tumor (human breast cancer transplanted into nude mice) [24]. Thus, AFP-derived GIP acting as a chemokine decoy ligand could possibly affect multiple phases of the development, oncogenic growth, and metastasis of tumors.

## 6. How Could GIP Treatment Impede the Metastatic Spread of Solid Tumors In Vivo? A Proposed Action

The AFP-derived peptide (GIP), when injected into the human/animal body, could follow a three-step course of action as now outlined: First, the in vivo administered (injected) peptide could gain entrance into the blood and/or interstitial fluid and travel toward and approach a cancer cell mass [60]. GIP has been reported to attach and/or adhere to the surface of the cell membrane (plasma membrane) of tumor cells [21]. By means of its pore forming or channel forming process, GIP has been observed to enter into the cell cytoplasmic compartment and distribute in a diffusive pattern [21]. The widely cytoplasmic-distributed GIP molecules then act to decrease the ATP energy reserve of the tumor cell [24]. After several minutes, GIP becomes localized as a thinly perinuclear layer surrounding the entire cell nucleus. In this manner, GIP might be in a position to selectively gate the transpassage traffic of molecules entering into and passing out of the nucleoplasm [21]. Theoretically, this gating might be possible to regulate the traffic flow of signals from cytoplasmic transcription factors to steroid and other receptors already residing in the nuclear compartment [61,62]. This step would also tend to cease the growth activities of the cell. By the same token, GIP might also engage and suppress the growth cycle in the cell cytoplasm by mechanisms already reported in the literature and cited above [5].

Secondly, GIP could make it more difficult for tumor cells to leave the original tumor mass. GIP is able to bind to the cell’s cytoskeletal fiber that furthermore allows the cell to change its shape. This action stiffens the skeletal frame of tumor cells so that detached tumor cells cannot easily traverse the blood vessels for passage (metastasize) to distal sites. Such cytoskeletal, cell shape, and tumor cell migration observations for GIP have been previously reported [20]. After entering into the blood vessels, GIP might block the ability of platelets to shield the circulating tumor cells from immune attacks and clearance from the blood stream as reported in GIP blood cell agglutination data [19,20].

Thirdly, GIP could block the ability of circulating cells to settle into distal homing sites as directed and determined by the homing agents of chemokines and their receptor binding partners. GIP is capable of blocking the signal transduction, attachment, and adherence to proteins of the extracellular matrix (ECM) residing in distal sites of newly metastatic nesting locations in various organs and tissues [19,20,21].

## 7. Conclusions

It can be gleaned from the studies described in this treatise that a peptide (GIP) derived from the full-length AFP protein molecule contains an occult peptide fragment that can inhibit or suppress cancer metastasis in both cell culture and in xenograft animal models [58]. The present results demonstrated that GIP was capable of suppressing or inhibiting both tumor growth progression and metastasis, via cell cycle arrest [5]. In addition, recent data implied that tumors presenting with transient receptor potential (TRP) calcium-regulated cation channels could potentially lead to cell death from calcium toxicity [40]. Finally, a proposed mechanism of action was presented regarding how GIP might be capable of inhibiting the metastatic spread of solid tumors throughout the body. Such treatment using an AFP-derived peptide (like GIP) might provide a non-toxic therapeutic treatment option which lacks side effects; thus, it could serve as an adjunct treatment modality to aid in chemotherapy and to block metastatic migration itself.

## Figures and Tables

**Table 1 cancers-16-01531-t001:** Global RNA microarray data from MCF-1 breast cancer cells treated with GIP. Transcripts displaying 1.0 or larger log-fold (log base 2.0) decrease for genes associated with cell division and proliferation processes, ubiquitination, and cation channels obtained from human MCF-7 breast cancer cell in vitro (see Ref. [18]).

Gene Title	Fold Decrease	Cell Function
I. Cell Cycle Regulation		
1. Calpain (LOC 441200)	−32.5	Cell cycle progression
2. F-Boc/Wd40, Domain-10 (FBXW10)	−14.9	P27 degradation pathway
3. Serine/Threonine Kinase-33 (STK33)	−9.2	SH3 protein kinase
4. Establishment of Cohesion-1, Homolog (ESC02)	−9.2	DNA replication
5. Checkpoint Suppressor-1 (CHES1) (FOXN3)	−9.2	S-phase checkpoint
6. Cyclin-E	−4.6	Regulates G-S transition
7. SKP2 **	−4.3	Mediates p27 degradation
8. Transcription Dp-1 (TFDP1)	−4.3	Binds E2F-1; G1 to S
9. CDC20 Cell Division Homolog	−4.3	Activates ubiquitin
10. Triple Function Domain (TRIO)	−3.7	Actin remodeling
11. Histone-1, H4g (HIST1H4G)	−3.2	DNA repair/replication
12. Fanconi Anemia-D2 (FRANCD2)	−2.0	DNA repair/synthesis
II. Channel Associated Proteins		
1. Potassium Voltage-gated Channel (KCNB2)	−8.0	Shab ion channel
2. Transmembrane Channel-Like 5 (TMC5)	−5.2	Ion transporter
3. Potassium Voltage-gated Channel, KQT-like (KCNQ3)	−4.0	Cation signaling
4. Calcium Channel, Voltage dependent of 2 (CACNA2D4)	−2.0	Calcium signaling
5. Calcium/Calmodulin-dependent Kinase (CAMK2B)	−1.9	Calcium regulation
6. Calcineurin A gamma (PPP3CC)	−1.8	Calcium phosphate 3 protein
7. Calcium Channel, Voltage Dependent (CACNC6)	−1.8	Calcium transport

**Table 2 cancers-16-01531-t002:** Amino acid sequencing matching of alpha-fetoprotein (AFP)-derived Growth Inhibitory Peptide (GIP, P149) with various cation channel-associated and calcium-interacting proteins including ion co-transporters.

		% Identity/Similarity	% Total
Hum GIP #445	L S E D K L L A C G E G A A D I I I G H L C I R H E M T P V N P G V G N	100/100	100
Fragment GIP	GIPa GIPb GIPc		
Hum Na, Glu COTR (#163)	L S G H V L R S C I H P A G S X G L E H L C L R H	42/38	80
Xen Na/K ATPase (#252)	L S C T R L I A C C Y G N C T G A I X H L C X X T N L S S I	36/23	59
Hum Actinin, α (#958)	L S E Q R L L P R G E G	62/23	85
Na Chanel Protein (#55)	Y V Q D Q L Q A C G E G	58/25	83
Hum CoFilin (#40)	L S E D K K N	71/0	71
Hum Calmodulin (#27)	L S E I E L L	71/0	71
Xen Ache R, δ (#362)	L S G D K L L S I	66/33	99
Yeast MDRP ($735)	L S E N K L L S P S	60/30	90
Mus Entactin (#1935)	K L L S C G E H	63/25	88
Rab Ca Channel-P (#32)	G L L P C A E G	63/25	88
Hum Nic- AcheR α (#440)	C G E V L R D V V F G L W C I R D K A T G G G S G	40/20	60
Rat Musc AcheR (#588)	C G N G P S R R I R A L D C L R L G R K S G A S G V G	33/41	74
Carp Ca Channel-P (#333)	L C G E G A A G L	33/11	44
Piso ATP-syn A (31165)	A A N L T A G H L L	45/45	90
Hum Calcitonin R (#210)	N S M I I I I H L C	50/30	80
Pig Calcitonin R (#195)	N S I I I I I H L V	50/30	80
Rat Calcitonin R (#195)	N S I I I I I H L V	50/30	70
Hum calreticulin (#3692)	I Q S I I V G H L G	50/20	70
Yeast Calmodulin (#21635)	N R I G Q L C I R	66/11	77
Pig Na/Glu COTR (#1150)	I I L S Q L C I R	56/33	89
Hum Calcitonin (#1550)	L C I R H S F T P A	60/30	90
Mus K-Chanel P (#18)	L C I R G T L T P R	60/20	80
Bov ATP-channel (P) (#385)	C I Q F E L P P V N	50/30	80
Rat Ca/ATPase (#660)	C I H N Q M Q P V H	60/40	100
Rat Ca/Calm Kin-α (#118)	C I H Q I L E S V N	40/50	90
Rat Ca/Calm Kin-D (#118)	C I Q Q I L E S V N	40/50	90
Rat Na/GLN/ASNtR (#228)	R I R E E M V P V P G S V	54/38	92
Caeel Mech-Sen (#332)	C I K H E H A A M V L N L W E	27/47	74
Pig Na/Glu COTR (#510)	T A Y K P S I G N	56/22	78

Legend: Hum = Human; Xen = Xenopus; Na = Sodium; Mus = Mouse; Carp = Fish; and Caeel = C. elegans.

**Table 3 cancers-16-01531-t003:** Sequence matching of conserved amino acids of chemokine-associated proteins with alpha-fetoprotein Growth Inhibitory Peptide—P149.

																																			% Identity/Similarity	% Total
AFP Peptide GIP	L	S	E	D	K	L	L	A	C	G	E	G	A	A	D	I	I	I	G	H	L	C	I	R	H	E	M	T	P	V	N	P	G	V	100/0	100
Fragments of GIP						G	I	P	a										G	I	P	b							G	I	P	c				
Mus IL-1b		Q	E	D	R	L	V	L	C	L	X	G	A	X	D	V	P	V	G	E	L	C	R	L	K										42/25	67
Mus Thy 1.2	L	S	T	D	K	L	V	K	C	G	G	I	S																						54/15	69
RatThy-1	L	S	T	D	K	L	V	K	C	G	G	I	S																						54/15	69
Rab CD1-1					K	L	L	P	C	G	L	S																							63/13	76
Mus IL-2 Receptor	L	A	S	X	G	L	L									I	P	E	G	H	L	C	I	L	N	D									50/11	61
Hum IFN α/β													I	Q	S	I	I	V	G	H	L	G													50/20	70
Hum IL-2													I	X	S	I	I	V	G	H	L	L	G												50/10	60
Hum CD-1													I	Q	S	I	I	V	G	H	L	G													50/20	70
Mus C5S															R	V	T	I	G	P	L	C	I	R											60/20	80
Mus pro-C5															R	V	T	I	G	P	L	C	T	R											60/20	80
Mus C5D															R	V	T	I	G	P	L	C	T	R											60/20	80
Hum IFN-γ																						C	I	S	I	S	N	Q	P	V	N	P			56/9	65
Hum Leuc CD-9																						C	I	Q	R	Q	V	P	P	V	X	P			45/27	72
Mus t-cell receptor																						C	I	R	D	N	K	T	P	S	T				50/20	70
Hum IgG H-chain																						C	I	H	H	S	L	T	P	P	D				50/30	80
SDF-1α	V	N	K	L	K	I	L	N	C													C	I	K	W	E	Y		K	L	N	K			33/50	83

Legend: Mus = Mouse; Hum = Human; Rab = rabbit; and H-chain = heavy chain.

**Table 4 cancers-16-01531-t004:** Comparisons between chemokines versus Growth Inhibitory Peptide (GIP).

List of Properties	Chemokines	Growth Inhibitory Peptide
(1) Amino acid length	60–90 Amino acids	34-amino acid (⅓ to ½’ the size)
(2) Molecular weight (Daltons)	8 to 10,000 Daltons	3573 Daltons
(3) Type of protein	Largely cationic and anionic fragment	Amphipathic
(4) Oligomer formation	Dimers, tetramers, monomers	Monomer, dimer, trimer, hexamers
(5) Effective concentration range	Nanomolar to low micromolar	Nanomolar to low micromolar
(6) Secondary structure	Antiparallel B strands; α helix at c terminal Random coil at NH_2_ COOH terminals	B strands and turns (45%); α helical only 10%, 45% random coil
(7) Receptor binding	Micromolar range	Micromolar range
(8) Hydrophobicity	Surface hydrophobicity; central hydrophobicity	Surface hydrophobicity (mod piece)
(9) Integrin response	Integrin activation and signaling	Integrin interaction and signaling
(10) Calcium (Ca^2+^) responsiveness	Induces Ca^2+^ mobilization	Sensitive Ca^2+^ responsiveness (CD and bioassay)
(11) Microtubule response	Induces actin polymerization	Induces tubulin polymerization
(12) Cancer growth	Inhibits growth of cancer without toxicity	Inhibits growth of cancer without toxicity
(13) Mouse model of diabetes (NOD)	Induces early onset of diabetes at high doses	Induces early onset of diabetes at high doses
(14) Species specificity	Present and reactive in rabbit, human, mouse, chicken, and frog (xenopus)	Present and reactive in human, mouse, chicken, rat, frog, and brine shrimp
(15) Tissue localization (SDF-1 as ligand)	Heart, brain, pancreas, placenta, lung, liver, muscle, kidney, spleen, thymus, prostate, testis, ovary, small intestine. Colon	Active against colon, ovary, breast, prostate, NSC lung, skin, central nervous system (brain, kidney, leukocytes, lymphocytes, and uterine tissues)
(16) Receptor type	G-coupled receptors CCR5, CXCR4	CCR5, CXCR4
(17) Platelet aggregation	Regulation of platelet aggregation	Inhibits platelet aggregation
(18) Cell migration	Regulates and modulates cell migration	Inhibits cell migration
(19) Cell adhesion	Inhibits/enhances cell adhesion	Inhibits/enhances cell adhesion
(20) Apoptosis, programmed cell death	Regulation of apoptosis (inhibition/enhancement)	Inhibits apoptosis; enhances α-irradiated apoptosis
(21) Angiogenesis; neovascularization	Regulation of angiogenesis (inhibition/enhancement)	Inhibits angiogenesis
(22) Increased vascular permeability; basement membrane disruption	Induces and modulates vascular permeability; induces ascites formation	Inhibits vascular permeability in ascites tumors
(23) Cell proliferation, tumor growth	Regulates cellular proliferation, cancer (enhancement/inhibition)	Inhibits cell proliferation of prostate/breast cancer and 9 tumor types
(24) Cellular differentiation	Regulates cellular differentiation, myelogenesis, lymphogenesis	Inhibits frog metamorphosis and fetal chick development
(25) Estrogen-regulated (induced) molecules	Chemokine CXCL12 (SDF-1) is estrogen-sensitive (induced) molecule	GIP-midpiece (p232) binds estradiol
(26) Estrogen (E2)-induced mitogenic activity	SDF-1 mediates proliferative action of (E2) estradiol	GIP suppresses proliferative action of (E2) estradiol
(27) Estrogen regulation mediated by estrogen receptor (Erα)	SDF-1 induction is mediated through Erα	GIP binds Erα (receptor)
(28) MAP kinase-mediated pathways	SDF-1 is upstream effector of MAP kinase	MAP kinase is down-regulated by GIP
(29) Use of chemokine CXCR4 receptor	CXCR4 is sole receptor of SDF-1	CXCR4 modulates growth in ovary and breast cancer which GIP suppresses
(30) Use of CCR5 chemokine receptor in HIV and use of CCR6	HIV binds to CCR5 co-receptor and to CD4 receptor, CCL28 binds CCR6	Human AFP (full-length) binds to CCR5 receptor

Legend: CD = circular dichroism; CXCL12 = chemokine ligand (SDF-1); CCR5, CXCR4 = chemokine receptors; and E2 = estradiol.

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
