# Peer review of "The Role of Ion Channels and Chemokines in Cancer Growth and Metastasis: A Proposed Mode of Action Using Peptides in Cancer Therapy"

_cancers, 2024, doi:10.3390/cancers16081531_

Round 1
Reviewer 1 Report
Comments and Suggestions for Authors
In this review Dr. Mizejewski revisited the relevance of specific peptides derived from key metastatic players such as chemokines and their receptors and ion channels as potential targets for therapeutics intervention against metastasis. The manuscript is well written and the author has done a wonderful job classifying these peptides and describing their potential roles as regulators of cell cycle, calcium channels and cell adherence, all key events in the metastatic process. This review is suitable and timely.
Comments on the Quality of English LanguageA well written manuscript
Author Response
The reviewer stated 3 grammatical errors involving the superscript of calcium (Ca2++) which were easily corrected on my revised manuscript . Next the Reviewer stated that my manuscript reviewed the relevance of peptides to the metastatic components of chemokines, their receptors, and ion channels. This was my intended purpose of the submitted paper. The reviewer next stated the fact that a peptide can regulate the cell cycle, calcium channels, and cell adherence. The reviewer culminated the review by stating that the paper was both suitable and timely.

Reviewer 2 Report
Comments and Suggestions for Authors
Summary:
The author reviewed the function of ion channels and chemokines in metastasis and how GIP works on metastasis through action on ion channels and chemokines.
Comments:
The manuscript could be smoother for readers if it can be better-organized.
1. For session 2, I suggest that the authors describe GIP following this order: origin, structure, function at the molecular level, action at the physiology and pathology level.
2. The title of this review does not include tumor cell proliferation. But in session 3, the authors wrote ion channels' impact on proliferation mixing with metastasis. I suggest that the authors write the impact of ion channels on cell cycle/proliferation and metastasis separately.
3. Session 4 is about Ca+ channels' effect on cell death not specific to cancer cells and it's hard to relate cell death with metastasis based on the authors narration.
4 For session 5, the logic of paragraph 3 (line 275-288) is not so clear. The evidence supposed to be from researches that GIP is serving as a chemokine ligand is not solid.
Author Response
Answers to Reviewer’s Comments:
Reviewer # 2:
The whole paper was better organized by a MAJOR rewrite and Revision.
Cell proliferation , growth, and metastasis were separated on pages 2 to 4.
Cal++ channels and cell death were revised on pages 8 to 10.
Evidence that GIP serves as a chemokine ligand is not solid. This subject was revised to suggest that GIP was only postulated (suggested) to serve as a chemokine ligand.

Reviewer 3 Report
Comments and Suggestions for Authors
Growth Inhibitory Peptide (GIP) was found to associate with cell membrane penetration and disruption, and can suppress cancer cell growth and migration. This manuscript focused on the regulation of ion channels, chemokines, and metastasis by GIP, and the potential peptide therapy for cancer treatment The topic is interesting. However, there are issues to be considered.
Minor concern:
l Line 100, “-45 mVolts” should be “-45 mV”.
l Line 141, 187. The “Ca++” should be “Ca2+”.
l Line 209. The “Ca” should not be superscript.
Major concern:
l Line 293. “The influx of calcium ions as a result of the TRP channel modulation (a low negative membrane potential) would cause an increase the cytoplasmic calcium levels. Thus, the cell which normally responds by pumping out calcium through some type of TRP channel.” Please specify and cite the references which TRP channels are responsible for pumping out of calcium. As far as my knowledge is concerned, Most of the TRP channels are non-selective cation channels, not ATP-driven Ca2+ pumps.
l It is still unclear the mechanisms of the regulation of K+ channel by GIP. By regulation of membrane potential? Or changes in intracellular calcium concentrations? Or others?
l Moreover, it is unclear whether GIP-K+ channel-mediated cell function regulation is involved in intracellular Ca2+ concentration changes or other signal transduction signaling?
l Most of TRP channels are ligand gated or store-operated calcium channels. It is unclear how GIP activates TRP channel to induce Ca2+ influx.
l Although the author cited the references to reveal that stimulation of BK channels by menthol increased the channel open probability and mean open frequency. The author implied that GIP may also activate TRP calcium channels. Please cite the specific references to demonstrate the activation of TRP channels by GIP. Please also discuss what types of TRP channels are involved in the GIP-mediated cell function regulation.
l Calcium signaling is reported to promote cancer metastasis. Whether GIP induces cancer cell cycle arrest and apoptosis rather than metastasis?
l Matrix metalloproteinases (MMPs) play important roles in facilitating the metastasis of tumor cells. GIP was capable of suppressing or inhibiting both tumor growth progression and metastasis. It is unclear whether MMPs are down-regulation or specific signaling pathways involved in metastasis are deactivated in the GIP-mediated inhibition?
l The author demonstrated the Global RNA microarray data from MCF-1 breast cancer cells treated with GIP in the manuscript. Please also discuss other types of cancers in the GIP-mediated cell functions regulation.
l Overall, the manuscript need to be well organized.
Comments on the Quality of English LanguageMinor editing of English language required.
Author Response
Answers to Reviewer’s Comments:
REVIEWER #3:
1) The influx of Ca+ would cause an increase of cytoplasmic calcium levels and References were added to signify this. Indeed, TRP channels are non-selective cation channels and not ATP driven.
2) The mechanism of GIP-K channel mediated regulation remains unclear and that is why I just postulated on the mechanisms.
3) TRP channels are ligand -gated or store-operated calcium channels and it was confirmed by the unpublished grant results by DR. Wondergem and Dr. Wonderlin (refer to the GRANT ACKNOWLEDGEMENTS at the end of the paper). Since it was unpublished results from a confidential BIOTECH GRANT, I only postulated the mechanism. Thus, references cannot be cited.
4) The same reply refers to #3 response above. The TRP channels involved are the TRP
Vallinoid-1, and the TRP Melastatins-- M1,M2, M7, and M8.see ref #41.-6 and #6 section.
5) Calcium signaling is reported to promote cancer metastasis .GIP has been published and reported to suppress cancer growth and metastasis in the References that I provided. GIP arrests growth by Cell Cycle G1 to S phase arrest and can suppress , reduce, or retard metastasis by the
Inhibition of platelet aggregation within blood vessels and the blockage of tumor cell adhesion to the Extracellular matrix and cells of endpoint nesting sites.
It is unclear whether MMPs are down-regulated or respond to specific signalling pathway
Deactivation. This subject was published in several of my papers in reference #6 and journal papers demonstrating that GIP acts like a disintegrin such as in : Journal:Cancer Studies , Vol-8(2): 1-7, 2023,and Internatl. Journal of cancer research & Mol. Mech. 2020, Vol-5(2):1 to 6 pages.
6) Please discuss other types of cancer growth suppressed and inhibited by GIP regulation.
A recent publication by the present author (GJM) demonstrates that GIP ca suppress nine different types of cancers (see Medical Research Archives , Vol -11(7):pages 1 to 15.

Round 2
Reviewer 2 Report
Comments and Suggestions for Authors
The manuscript is well organized and sufficiently improved.
Author Response
All of the comments were addressed by the author.

Reviewer 3 Report
Comments and Suggestions for Authors
Although the author revised the manuscript, the quality of manuscript was not significantly improved. I suggest the authors re-organize the manuscript and re-submit again.
l References must be numbered in order of appearance in the text (including table captions).
l According to the role of author guideline of IJMS. No new, unpublished data should be presented in the review article.
l Please cite the specific references for “GIP at low concentrations could serve as a “cell channel forming” peptide coincident with a decreased cell membrane electrical resistance” and “GIP was capable of cell membrane pore forming activities with a cell membrane potential at -30 to 133 -45 mV”. The references are not relevant to the contents of the manuscript. The personal communication or unpublished data should not be cited in review article.
The descriptions from reference #25: “In vivo recordings indicated that GIP-34 at 10-6M and lower concentration could serve as cell membrane pore forming/channel- interacting peptide coincident with a decreased cell membrane resistance. At higher peptide concentrations (10-5M and greater) GIP-34 was capable of membrane penetration while stabilizing the cell membrane potential at -30 to -45 mVolts [42]. GIP-34 could also act as a channel-interacting (or blocking) peptide coincident with increased cell membrane resistance”.
As description from reference #25, the GIP could act as channel-interacting peptide, not serve as a “channel forming” peptide.
l Please cite the specific references for table 1. The results of global RNA microarray data from “MCF-1” or “MCF-7” breast cancer cells treated with GIP could not be found in reference #18.
l Again, please cite the specific references for table 2-4.
Comments on the Quality of English LanguageMinor editing of English language required.
Author Response
#1) References should numbered in the text appearances . ANSWER: This has been done! #2)No new unpublished data should be presented in a review paper. ANSWER: The title of my paper is not a REVIEW PAPER: rather it is titled as a "PROPOSED MODE OF ACTION" paper which is a commentary type paper. #2)Cite the Reference for GIP as a pore-forming peptide. ANSWER: The Reference #5 describes this event on pages 2712 and in Figure-3.CANCERS, 2011, vol-3,pages 2709-2733.REF #25 was a published paper by the present author and ddi not need to be re-reviewed by REVIEWER #3. #4) Descriptions from Reference #25 is again ibacked up from Ref #5 showing that GIP can serve either as a pore-forming or channel forming / interactor peptide as stated in my earlier paper on page 2725. From Reference #5. #5)Cite the specific reference for an RNA MICROARRAY analysis of channel proteins from MCF-7 Cultured Human breast cancer. ANSWER : The global RNA microarray data could be found on page 2717 from my published paper in CANCERS, 2011, vol-3, pages 2709-2733. #6) Cite the specific references for TABLES 2 to 4. ANSWER: These TABLES have no references to Cite because they were generated from my own computer programs in my research laboratory At the WADSWORTH RESEARCH INSTITUTE, NEW YORK STATE DEPT. of HEALTH, Albany , N. Y. #7) MINOR editing of the ENGLISH Language may be done. ANSWER: This was done.
